# DIVA: Domain Invariant Variational Autoencoders

## Abstract

We consider the problem of domain generalization, namely, how to learn representations given data from a set of domains that generalize to data from a previously unseen domain. We propose the Domain Invariant Variational Autoencoder (DIVA), a generative model that tackles this problem by learning three independent latent subspaces, one for the domain, one for the class, and one for any residual variations. We highlight that due to the generative nature of our model we can also incorporate unlabeled data from known or previously unseen domains. To the best of our knowledge this has not been done before in a domain generalization setting. This property is highly desirable in fields like medical imaging where labeled data is scarce. We experimentally evaluate our model on the rotated MNIST benchmark and a malaria cell images dataset where we show that (i) the learned subspaces are indeed complementary to each other, (ii) we improve upon recent works on this task and (iii) incorporating unlabelled data can boost the performance even further.

## 1 Introduction

Deep neural networks (DNNs) led to major breakthroughs in a variety of areas like computer vision and natural language processing. Despite their big success, recent research shows that DNNs learn the bias present in the training data. As a result they are not invariant to cues that are irrelevant to the actual task (Azulay & Weiss, 2018). This leads to a dramatic performance decrease when tested on data from a different distribution with a different bias.

In domain generalization the goal is to learn representations from a set of similar distributions, here called domains, that can be transferred to a previously unseen domain during test time. A common motivating application, where domain generalization is crucial, is medical imaging (Blanchard et al., 2011; Muandet et al., 2013). For instance, in digital histopathology a typical task is the classification of benign and malignant tissue. However, the preparation of a histopathology image includes the staining and scanning of tissue which can greatly vary between hospitals. Moreover, a sample from a patient could be preserved in different conditions (Ciompi et al., 2017). As a result, each patient's data could be treated as a separate domain (Lafarge et al., 2017). Another problem commonly encountered in medical imaging is class label scarcity. Annotating medical images is an extremely time consuming task that requires expert knowledge. However, obtaining domain labels is surprisingly cheap, since hospitals generally store information about the patient (e.g., age and sex) and the medical equipment (e.g., manufacturer and settings). Therefore, we are interested in extending the domain generalization framework to be able to deal with additional unlabeled data, as we hypothesize that it can help to improve performance.

In this paper, we propose to tackle domain generalization via a new deep generative model that we refer to as the Domain Invariant Variational Autoencoder (DIVA). We extend the variational autoencoder (VAE) framework (Kingma & Welling, 2013; Rezende et al., 2014) by introducing independent latent representations for a domain label, a class label and any residual variations in the input $\mathbf{x}$. Such partitioning of the latent space will encourage and guide the model to disentangle these sources of variation. Finally, by virtue of having a generative model we can naturally handle the semi-supervised scenario, similarly to Kingma et al. (2014). We evaluate our model on a version of the MNIST dataset where each domain corresponds to a specific rotation angle of the digits, as well as on a Malaria Cell Images dataset where each domain corresponds to a different patient. An implementation of DIVA can be found under (URL was removed to preserve anonymity).

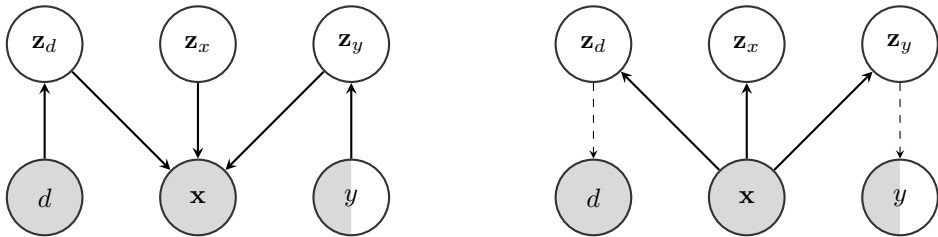

Figure 1: Left: Generative model. According to the graphical model we obtain $p(d, \mathbf{x}, y, \mathbf{z}_d, \mathbf{z}_x, \mathbf{z}_y) = p_\theta(\mathbf{x}|\mathbf{z}_d, \mathbf{z}_x, \mathbf{z}_y)p_{\theta_d}(\mathbf{z}_d|d)p(\mathbf{z}_x)p_{\theta_y}(\mathbf{z}_y|y)p(d)p(y)$. Right: Inference model. We propose to factorize the variational posterior as $q_{\phi_d}(\mathbf{z}_d|\mathbf{x})q_{\phi_x}(\mathbf{z}_x|\mathbf{x})q_{\phi_y}(\mathbf{z}_y|\mathbf{x})$. Dashed arrows represent the two auxiliary classifiers $q_{\omega_d}(d|\mathbf{z}_d)$ and $q_{\omega_y}(y|\mathbf{z}_y)$.

## 2 TOWARDS DOMAIN GENERALIZATION WITH GENERATIVE MODELS

We follow the domain generalization definitions used in Muandet et al. (2013). A domain is defined as a joint distribution $p(\mathbf{x}, y)$ on $\mathcal{X} \times \mathcal{Y}$, where $\mathcal{X}$ denotes the input space and $\mathcal{Y}$ denotes the output space. Let $\mathfrak{P}_{\mathcal{X} \times \mathcal{Y}}$ be the set of all domains. The training set consists of samples $\mathcal{S}$ taken from $N$ domains, $\mathcal{S} = \{S^{(d=i)}\}_{i=1}^N$. Here, the $i$th domain $p^{(d=i)}(\mathbf{x}, y)$ is represented by $n_i$ samples, $S^{(d=i)} = \{(\mathbf{x}_k^{(d=i)}, y_k^{(d=i)})\}_{k=1}^{n_i}$. Each of the $N$ distributions $p^{(d=1)}(\mathbf{x}, y), \dots, p^{(d=i)}(\mathbf{x}, y), \dots, p^{(d=N)}(\mathbf{x}, y)$ are sampled from $\mathfrak{P}_{\mathcal{X} \times \mathcal{Y}}$. We further assume that $p^{(d=i)}(\mathbf{x}, y) \neq p^{(d=j)}(\mathbf{x}, y)$, therefore, the samples in $\mathcal{S}$ are non-i.i.d. During test time we are presented with samples $S^{(d=N+1)}$ from a previously unseen domain $p^{(d=N+1)}(\mathbf{x}, y)$. We are interested in learning representations that generalize from $p^{(d=1)}(\mathbf{x}, y), \dots, p^{(d=N)}(\mathbf{x}, y)$ to this new domain. Training data are given as tuples $(d, \mathbf{x}, y)$ in the case of supervised data or as $(d, \mathbf{x})$ in the case of unsupervised data.

### 2.1 DIVA: DOMAIN INVARIANT VAE

Assuming a perfectly disentangled latent space (Higgins et al., 2018), we hypothesize that there exists a latent subspace that is invariant to changes in $d$, i.e., it is domain invariant. We propose a generative model with three independent sources of variation; $\mathbf{z}_d$, which is domain specific, $\mathbf{z}_y$, which is class specific and finally $\mathbf{z}_x$, which captures any residual variations left in $\mathbf{x}$. While $\mathbf{z}_x$ keeps an independent Gaussian prior $p(\mathbf{z}_x)$, $\mathbf{z}_d$ and $\mathbf{z}_y$ have conditional priors $p_{\theta_d}(\mathbf{z}_d|d)$, $p_{\theta_y}(\mathbf{z}_y|y)$ with learnable parameters $\theta_d, \theta_y$. This will encourage information about the domain $d$ and label $y$ to be encoded into $\mathbf{z}_d$ and $\mathbf{z}_y$, respectively. Furthermore, as $\mathbf{z}_d$ and $\mathbf{z}_y$ are marginally independent by construction, we argue that the model will learn representations $\mathbf{z}_y$ that are invariant with respect to the domain $d$. All three of these latent variables are then used by a single decoder $p_\theta(\mathbf{x}|\mathbf{z}_d, \mathbf{z}_x, \mathbf{z}_y)$ for the reconstruction of $\mathbf{x}$.

Since we are interested in using neural networks to parameterize $p_\theta(\mathbf{x}|\mathbf{z}_d, \mathbf{z}_x, \mathbf{z}_y)$, exact inference will be intractable. For this reason, we perform amortized variational inference with an inference network (Kingma & Welling, 2013; Rezende et al., 2014), i.e., we employ a VAE-type framework. We introduce three separate encoders $q_{\phi_d}(\mathbf{z}_d|\mathbf{x})$, $q_{\phi_x}(\mathbf{z}_x|\mathbf{x})$ and $q_{\phi_y}(\mathbf{z}_y|\mathbf{x})$ that serve as variational posteriors over the latent variables. Notice that we do not share their parameters as we empirically found that sharing parameters leads to a decreased generalization performance. For the prior and variational posterior distributions over the latent variables $\mathbf{z}_x, \mathbf{z}_d, \mathbf{z}_y$ we assume fully factorized Gaussians with parameters given as a function of their input. We coin the term Domain Invariant VAE (DIVA) for our overall model, which is depicted in Figure 1.

Given a specific dataset, all of the aforementioned parameters can be optimized by maximizing the following variational lower bound per input $\mathbf{x}$:

$$\mathcal{L}_s(d, \mathbf{x}, y) = \mathbb{E}_{q_{\phi_d}(\mathbf{z}_d|\mathbf{x})q_{\phi_x}(\mathbf{z}_x|\mathbf{x}), q_{\phi_y}(\mathbf{z}_y|\mathbf{x})} \left[ \log p_\theta(\mathbf{x}|\mathbf{z}_d, \mathbf{z}_x, \mathbf{z}_y) \right]$$
$$- \beta KL\left( q_{\phi_d}(\mathbf{z}_d|\mathbf{x})||p_{\theta_d}(\mathbf{z}_d|d) \right) - \beta KL\left( q_{\phi_x}(\mathbf{z}_x|\mathbf{x})||p(\mathbf{z}_x) \right)$$
$$- \beta KL\left( q_{\phi_y}(\mathbf{z}_y|\mathbf{x})||p_{\theta_y}(\mathbf{z}_y|y) \right). \tag{1}$$

Notice that we have introduced a weigting term, $\beta$. This is motivated by the $\beta$-VAE (Higgins et al., 2017) and serves as a constraint that controls the capacity of the latent spaces of DIVA. Larger

values of $\beta$ limit the capacity of each $\mathbf{z}$ and in the ideal case each dimension of $\mathbf{z}$ captures one of the conditionally independent factors in $\mathbf{x}$.

To further encourage separation of $\mathbf{z}_d$ and $\mathbf{z}_y$ into domain and class specific information respectively, we add two auxiliary objectives. During training $\mathbf{z}_d$ is used to predict the domain $d$ and $\mathbf{z}_y$ is used to predict the class $y$ for a given input $\mathbf{x}$:

$$\mathcal{F}_{\text{DIVA}}(d, \mathbf{x}, y) := \mathcal{L}_s(d, \mathbf{x}, y) + \alpha_d \mathbb{E}_{q_{\phi_d}(\mathbf{z}_d | \mathbf{x})} \left[ \log q_{\omega_d}(d | \mathbf{z}_d) \right] + \alpha_y \mathbb{E}_{q_{\phi_y}(\mathbf{z}_y | \mathbf{x})} \left[ \log q_{\omega_y}(y | \mathbf{z}_y) \right],$$
(2)

where $\alpha_d$, $\alpha_y$ are weighting terms for each of these auxiliary objectives. Since our main goal is a domain invariant classifier, during inference we only use the encoder $q_{\phi_y}(\mathbf{z}_y | \mathbf{x})$ and the auxiliary classifier $q_{\omega_y}(y | \mathbf{z}_y)$. For the prediction of the class $y$ for a new input $x$ we use the mean of $\mathbf{z}_y$. Consequently, we regard the the variational lower bound $\mathcal{L}_s(d, \mathbf{x}, y)$ as a regularizer. Therefore, evaluating the marginal likelihood $p(\mathbf{x})$ of DIVA is outside the scope of this paper.

Locatello et al. (2018) and Dai & Wipf (2019) claim that learning a disentangled representation, i.e., $q_\phi(\mathbf{z}) = \prod_i q_\phi(z_i)$, in an unsupervised fashion is impossible for arbitrary generative models. Inductive biases, e.g., some form of supervision or constraints on the latent space, are necessary to find a specific set of solutions that matches the true generative model. Consequently, DIVA is using domain labels $d$ and class labels $y$ in addition to input data $\mathbf{x}$ during training. Recent work by Khemakhem et al. (2019) shows that conditional priors, like $p_{\theta_d}(z_d | d)$ and $p_{\theta_y}(z_y | y)$ in DIVA, lead to identifiability guarantees in VAEs. Furthermore, we enforce the factorization of the marginal distribution of $\mathbf{z}$ in the following form: $q_\phi(\mathbf{z}) = q_{\phi_d}(\mathbf{z}_d) q_{\phi_x}(\mathbf{z}_x) q_{\phi_y}(\mathbf{z}_y)$, which prevents the impossibility described in Locatello et al. (2018). We argue that the strong inductive biases in DIVA make it possible to learn disentangled representations that match the ground truth factors of interest, namely, the domain factors $\mathbf{z}_d$ and class factors $\mathbf{z}_y$. To highlight the importance of a partitioned latent space we compare DIVA to a VAE with a single latent space, the results of this comparison can be found in the Appendix.

## 2.2 SEMI-SUPERVISED DIVA

In Kingma et al. (2014) an extension to the VAE framework was introduced that allows to use labeled as well as unlabeled data during training. While Kingma et al. (2014) introduced a two step procedure, Louizos et al. (2015) presented a way of optimizing the decoder of the VAE and the auxiliary classifier jointly. We use the latter approach to learn from supervised data $\{(d_n, \mathbf{x}_n, y_n)\}$ as well as from unsupervised data $\{(d_m, \mathbf{x}_m)\}$. Analogically to Louizos et al. (2015), we use $q_{\omega_y}(y | \mathbf{z}_y)$ to impute $y$:

$$\begin{aligned}
\mathcal{L}_u(d, \mathbf{x}) = {} & \mathbb{E}_{q_{\phi_d}(\mathbf{z}_d | \mathbf{x}) q_{\phi_x}(\mathbf{z}_x | \mathbf{x}) q_{\phi_y}(\mathbf{z}_y | \mathbf{x})} [\log p_\theta(\mathbf{x} | \mathbf{z}_d, \mathbf{z}_x, \mathbf{z}_y)] \\
& - \beta KL(q_{\phi_d}(\mathbf{z}_d | \mathbf{x}) || p_{\theta_d}(\mathbf{z}_d | d)) - \beta KL(q_{\phi_x}(\mathbf{z}_x | \mathbf{x}) || p(\mathbf{z}_x)) \\
& + \beta \mathbb{E}_{q_{\phi_y}(\mathbf{z}_y | \mathbf{x}) q_{\omega_y}(y | \mathbf{z}_y)} [\log p_{\theta_y}(\mathbf{z}_y | y) - \log q_{\phi_y}(\mathbf{z}_y | \mathbf{x})] \\
& + \mathbb{E}_{q_{\phi_y}(\mathbf{z}_y | \mathbf{x}) q_{\omega_y}(y | \mathbf{z}_y)} [\log p(y) - \log q_{\omega_y}(y | \mathbf{z}_y)],
\end{aligned}$$
(3)

where we use Monte Carlo sampling with the reparametrization trick (Kingma & Welling, 2013) for the continuous latents $\mathbf{z}_d, \mathbf{z}_x, \mathbf{z}_y$ and explicitly marginalize over the discrete variable $y$. The final objective combines the supervised and unsupervised variational lower bound as well as the two auxiliary losses. Assuming $N$ labeled and $M$ unlabeled examples, we obtain the following objective:

$$\mathcal{F}_{\text{SS-DIVA}} = \sum_{n=1}^{N} \mathcal{F}_{\text{DIVA}}(\mathbf{x}_n, y_n, d_n) + \sum_{m=1}^{M} \mathcal{L}_u(\mathbf{x}_m, d_m) + \alpha_d \mathbb{E}_{q_{\phi_d}(\mathbf{z}_d | \mathbf{x}_m)} [\log q_{\omega_d}(d_m | \mathbf{z}_d)]. \quad (4)$$

## 3 RELATED WORK

The majority of proposed deep learning methods for domain generalization fall into one of two categories: 1) Learning a single domain invariant representation, e.g., using adversarial methods (Carlucci et al., 2018; Ghifary et al., 2015; Li et al., 2018; 2017; Motiian et al., 2017; Shankar et al., 2018; Wang et al., 2019). While DIVA falls under this category there is a key difference: we do not explicitly regularize $\mathbf{z}_y$ using $d$. Instead we learn complementary representations $\mathbf{z}_d$, $\mathbf{z}_x$ and $\mathbf{z}_y$ utilizing a generative architecture. 2) Ensembling models, each trained on an individual domain from

the training set (Ding & Fu, 2018; Mancini et al., 2018). The size of models in this category scales linearly with the amount of training domains. This leads to slow inference if the number of training domains is large. However, the size of DIVA is independent of the number of training domains. In addition, during inference time we only use the mean of the encoder $q_{\phi_y}(\mathbf{z}_y|\mathbf{x})$ and the auxiliary classifier $q_{\omega_y}(y|\mathbf{z}_y)$.

Concurrently to DIVA Cai et al. (2019) developed a framework to learn latent disentangled semantic representations (DSR) for domain adaptation. DSR assumes that the data generation process is exclusively controlled by the domain $d$ and class $y$. As a result, DSR is lacking a third latent space $z_x$. We designed DIVA assuming that not all variations in $x$ can be explained by the domain $d$ and the class $y$. Therefore we introduce $z_x$ in order to capture these residual variations. Furthermore, while DSR uses gradient reversal layers, we directly parameterize the ground truth generative model. As a result, the priors in DIVA are conditional which is necessary for guaranteed disentanglement as recent research has shown (Khemakhem et al., 2019). More related work is published under the name of multiple source domain adaptation (Zhao et al., 2018).

An area that is closely related to domain generalization is that of the statistical parity in fairness. The goal of fair classification is to learn a meaningful representation that at the same time cannot be used to associate a data sample to a certain group (Zemel et al., 2013). The major difference to domain generalization is the intention behind that goal, e.g., to protect groups of individuals *vs.* being robust to variations in the input. Consequently, DIVA is closely related to the fair VAE (Louizos et al., 2015). In contrast to the fair VAE, which is using a hierarchical latent space, DIVA is using a partitioned latent space. Moreover, the fair VAE requires the domain label during inference while DIVA alleviates this issue by learning the classifier without $d$. Similar to DIVA, there is an increasing number of methods showing the benefits of using latent subspaces in generative models (Siddharth et al., 2017; Klys et al., 2018; Jacobsen et al., 2018; Bouchacourt et al., 2018; Atanov et al., 2019).

We derived DIVA by following the VAE framework (Kingma & Welling, 2013), where the generative process is the starting point. A conditional version of the variational information bottleneck (CVIB) was proposed by Moyer et al. (2018) that likewise leads to an objective consisting of a reconstruction loss. However, CVIB suffers from the same limitation as the fair VAE: that the domain must be known during inference, hence, we excluded it from our experiments.

## 4 EXPERIMENTS

We evaluate the performance of DIVA on two datasets: Rotated MNIST (Ghifary et al., 2015) and Malaria Cell Images (Rajaraman et al., 2018). In both cases we first investigate if DIVA is able to successfully learn disentangled representations. Furthermore, we compare DIVA to other methods in a supervised and semi-supervised setting. While for the rotated MNIST dataset DIVA's graphical model is matching the ground truth generative model, the Malaria Cell Images dataset poses a more challenging and realistic scenario, where the ground truth generative model is unknown.

### 4.1 ROTATED MNIST

The construction of the Rotated MNIST dataset follows (Ghifary et al., 2015). We sample 100 images from each of the 10 classes from the original MNIST training dataset. This set of images is denoted $\mathcal{M}_{0^\circ}$. To create five additional domains the images in $\mathcal{M}_{0^\circ}$ are rotated by 15, 30, 45, 60 and 75 degrees. In order to evaluate their domain generalization abilities, models are trained on five domains and tested on the remaining 6th domain, e.g., train on $\mathcal{M}_{0^\circ}$, $\mathcal{M}_{15^\circ}$, $\mathcal{M}_{30^\circ}$, $\mathcal{M}_{45^\circ}$ and $\mathcal{M}_{60^\circ}$, test on $\mathcal{M}_{75^\circ}$. The evaluation metric is the classification accuracy on the test domain. All experiments are repeated 10 times. Detailed information about hyperparameters, architecture and training schedule can be found in the Appendix.

#### 4.1.1 QUALITATIVE DISENTANGLEMENT

First of all, we visualize the three latent spaces $\mathbf{z}_d$, $\mathbf{z}_x$ and $\mathbf{z}_y$, to see if DIVA is able to successfully disentangle them. In addition, we want to verify whether DIVA utilizes $\mathbf{z}_x$ in a meaningful way, since it is not directly connected to any downstream task. For the following visualizations we restrict the size of each latent space $\mathbf{z}_d$, $\mathbf{z}_x$ and $\mathbf{z}_y$ to 2 dimensions. Therefore, we can plot the latent subspaces

directly without applying dimensionality reduction, see Figure 2. Here, we trained DIVA on 5000 images from five domains: $\mathcal{M}_{0°}$, $\mathcal{M}_{15°}$, $\mathcal{M}_{30°}$, $\mathcal{M}_{45°}$ and $\mathcal{M}_{60°}$.

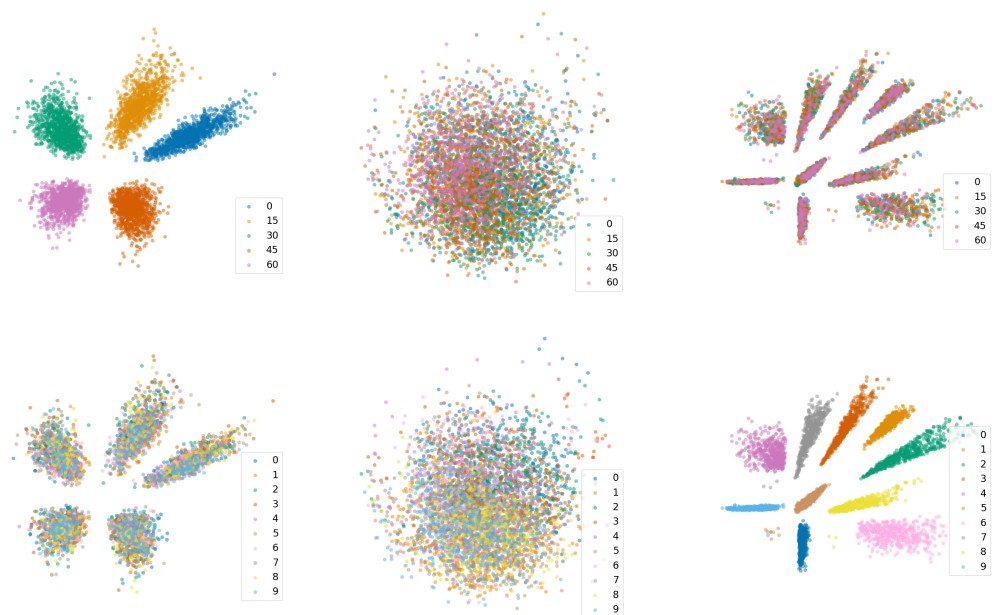

Figure 2: 2D embeddings of all three latent subspaces. In the top row embeddings are colored according to their domain, in the bottom row they are colored according to their class. First column: $\mathbf{z}_d$ encoded by $q_{\phi_d}(\mathbf{z}_d|\mathbf{x})$. The top plot shows five distinct clusters, where each cluster corresponds to a single domain. In the bottom plot no clustering is visible. Second column: $\mathbf{z}_x$ encoded by $q_{\phi_x}(\mathbf{z}_x|\mathbf{x})$. We observe a correlation between the rotation angle of each MNIST digit and $\mathbf{z}_x[0]$ in the top plot. Upon visual inspection of the original inputs $\mathbf{x}$, we find a correlation between the line thickness digit and $\mathbf{z}_x[0]$ as well as a correlation between the digit width and $\mathbf{z}_x[1]$ in the bottom plot. As a result, we observe a clustering of embeddings with class '1' at the lower left part of the plot. Third column: $\mathbf{z}_y$ encoded by $q_{\phi_y}(\mathbf{z}_y|\mathbf{x})$. In the top plot no clustering is visible. The bottom plot shows ten distinct clusters, where each cluster corresponds to a class. A plot of the 2D embeddings for the test domain $\mathcal{M}_{75°}$ can be found in the Appendix.

From these initial qualitative results we conclude that DIVA is disentangling the information contained in $\mathbf{x}$ as intended, as $\mathbf{z}_d$ is only containing information about $d$ and $\mathbf{z}_y$ only information about $y$. In the case of the Rotated MNIST dataset $\mathbf{z}_x$ captures any residual variation that is not explained by the domain $d$ or the class $y$. In addition, we are able to generate conditional reconstructions as well as entirely new samples with DIVA. We provide these in the Appendix.

### 4.1.2 COMPARISON TO OTHER METHODS

We compare DIVA against the well known domain adversarial neural networks (DA) (Ganin et al., 2015) as well as three recently proposed methods: LG (Shankar et al., 2018), HEX (Wang et al., 2019) and ADV (Wang et al., 2019). For the first half of Table 1 (until the vertical line) we only use labeled data. The first column indicates the rotation angle of the test domain. We report test accuracy on $y$ for all methods. For DIVA we report the mean and standard error for 10 repetitions. DIVA achieves the highest accuracy across all test domains and the highest average test accuracy among all proposed methods.

The second half of Table 1 showcases the ability of DIVA to use unlabeled data. For this experiment we add: The same amount (+1) of unlabeled data as well as three (+3), five (+5) and nine (+9) times the amount of unlabeled data to our training set. We first add the unlabeled data to $\mathcal{M}_{0°}$ and create the data for the other domains as described in Section 4.1. In Table 1 we can clearly see a performance increase when unlabeled data is added to the training set. When the amount of unlabeled data is

Table 1: Comparison with other state-of-the-art domain generalization methods. Methods in the first half of the table (until the vertical line) use only labeled data. The second half of the table shows results of DIVA when trained semi-supervised (+ X times the amount of unlabeled data). We report the average and standard error of the classification accuracy.

| Test | DA | LG | HEX | ADV | DIVA | DIVA(+1) | DIVA(+3) | DIVA(+5) | DIVA(+9) |
|------|------|------|------|------|------|------|------|------|------|
| $\mathcal{M}_{0°}$ | 86.7 | 89.7 | 90.1 | 89.9 | $\mathbf{93.5 \pm 0.3}$ | $93.8 \pm 0.4$ | $93.9 \pm 0.5$ | $93.2 \pm 0.5$ | $93.0 \pm 0.4$ |
| $\mathcal{M}_{15°}$ | 98.0 | 97.8 | 98.9 | 98.6 | $\mathbf{99.3 \pm 0.1}$ | $99.4 \pm 0.1$ | $99.5 \pm 0.1$ | $99.5 \pm 0.1$ | $99.6 \pm 0.1$ |
| $\mathcal{M}_{30°}$ | 97.8 | 98.0 | 98.9 | 98.8 | $\mathbf{99.1 \pm 0.1}$ | $99.3 \pm 0.1$ | $99.3 \pm 0.1$ | $99.3 \pm 0.1$ | $99.3 \pm 0.1$ |
| $\mathcal{M}_{45°}$ | 97.4 | 97.1 | 98.8 | 98.7 | $\mathbf{99.2 \pm 0.1}$ | $99.0 \pm 0.2$ | $99.2 \pm 0.1$ | $99.3 \pm 0.1$ | $99.3 \pm 0.1$ |
| $\mathcal{M}_{60°}$ | 96.9 | 96.6 | 98.3 | 98.6 | $\mathbf{99.3 \pm 0.1}$ | $99.4 \pm 0.1$ | $99.4 \pm 0.1$ | $99.4 \pm 0.1$ | $99.2 \pm 0.2$ |
| $\mathcal{M}_{75°}$ | 89.1 | 92.1 | 90.0 | 90.4 | $\mathbf{93.0 \pm 0.4}$ | $93.8 \pm 0.4$ | $93.8 \pm 0.2$ | $93.5 \pm 0.4$ | $93.2 \pm 0.3$ |
| Avg | 94.3 | 95.3 | 95.8 | 95.2 | $\mathbf{97.2 \pm 1.3}$ | $97.5 \pm 1.1$ | $97.5 \pm 1.2$ | $97.4 \pm 1.3$ | $97.3 \pm 1.3$ |

much larger than the amount of labeled data the balancing of loss terms become increasingly more challenging which can lead to a decling performance of DIVA, as seen in the last two columns of Table 1.

### 4.1.3 ADDITIONAL UNLABELED DOMAINS

In Section 4.1.2 we show that the performance of DIVA increases when it is presented with additional unlabeled data for each domain. As a result each training domain consists of labeled and unlabeled examples. In this section we investigate a more challenging scenario: We add an additional domain to our training set that consists of only unlabeled examples. Coming back to our introductory example of medical imaging, here we would add unlabeled data from a new patient or new hospital to the training set. This is in contrast to the experiment in Section 4.1.2 where we would add unlabeled data from each known patient or hospital to the training set.

In the following, we are looking at two different experimental setups, in both cases $\mathcal{M}_{75°}$ is the test domain: For the first experiment we choose the domains $\mathcal{M}_{0°}$, $\mathcal{M}_{15°}$, $\mathcal{M}_{45°}$ and $\mathcal{M}_{60°}$ to be part of the labeled training set. In addition, unlabeled data from $\mathcal{M}_{30°}$ is used. In Table 2 we can see that even in the case where the additional domain is dissimilar to the test domain DIVA is able to slightly improve. For the second experiment we choose the domains $\mathcal{M}_{0°}$, $\mathcal{M}_{15°}$, $\mathcal{M}_{30°}$ and $\mathcal{M}_{45°}$ to be part of the labeled training set. In addition, unlabeled data from $\mathcal{M}_{60°}$ is used. When comparing the results in Table 2 to the results in Table 1 we notice a drop in accuracy of about 20% for DIVA trained with only labeled data. However, when trained with unlabeled data from $\mathcal{M}_{60°}$ we see an improvement of about 7%. The comparison shows that DIVA can successfully learn from samples of a domain without any labels.

Table 2: Comparison of DIVA trained supervised to DIVA trained semi-supervised with additional unlabeled data from $\mathcal{M}_{30°}$ and $\mathcal{M}_{60°}$. We report the average and standard error of the classification accuracy on $\mathcal{M}_{75°}$.

| Unsupervised domain | DIVA supervised | DIVA semi-supervised |
|------|------|------|
| $\mathcal{M}_{30°}$ | $93.1 \pm 0.5$ | $93.3 \pm 0.4$ |
| $\mathcal{M}_{60°}$ | $73.8 \pm 0.8$ | $80.6 \pm 1.1$ |

### 4.2 MALARIA CELL IMAGES

The majority of medical imaging datasets consist of images from a multitude of patients. In a domain generalization setting each patient is viewed as an individual domain. While we focus on *patients as domains* in this paper, this type of reasoning can be extended to, e.g., *hospitals as domains*. We, among others (Rajaraman et al., 2018; Lafarge et al., 2017), argue that machine learning algorithms trained with medical imaging datasets should be evaluated on a subset of hold-out patients. This presents a more realistic scenario since the algorithm is tested on images from a previously unseen domain. In the following, we use a Malaria Cell Images dataset (Rajaraman et al., 2018) as an example of a dataset consisting of samples from multiple patients. The images in this dataset were collected and photographed at Chittagong Medical College Hospital, Bangladesh. It consists of

27558 single red blood cell images taken from 150 infected and 50 healthy patients. The images were manually annotated by a human expert. A cell has the label $y = 1$ if it shows the parasite and the label $y = 0$ if not. To facilitate the counting of parasitized and uninfected cells, the cells were stained using Giemsa stain which turns the parasites inside the cell pink. In addition, the staining process leads to a variety of colors of the cell itself. While the color of the cell is relatively constant for a single patient, it can vary greatly between patients, see the first row in Figure 3. This variability in appearance of the cells can be easily ignored by a human observer, however, machine learning models can fail to generalize across patients. In our experiments we will use the patient ID as the domain label $d$. We argue that for this specific dataset the patient ID is a good proxy of appearance variability. In addition, there is no extra cost for obtaining the patient ID for each cell.

Subsequently, we use a subset of the Malaria Cell Images dataset that consists of the 10 patients with the highest amount of cells. The amount of cells per patient varies between 400 and 700 and there are 5922 cell images in total. The choice of this subset is motivated by the similiar amount of cells as well as the similar marginal label distributions per patient, the latter being a necessary condition for successful domain generalization (Zhao et al., 2019). Furthermore we rescale all images to $64 \times 64$ pixels. To artificially expand the size of the training dataset we use data augmentation in the form of vertical flips, horizontal flips and random rotations.

### 4.2.1 QUALITATIVE DISENTANGLEMENT

We investigate the three latent subspaces $\mathbf{z}_d$, $\mathbf{z}_x$ and $\mathbf{z}_y$ to see if DIVA is able to successfully disentangle them. In addition, we want to see if DIVA utilizes $\mathbf{z}_x$ in a meaningful way, since it is not directly connected to any downstream task. Figure 3 shows the reconstructions of $\mathbf{x}$ using all three latent subspaces as well as reconstructions of $\mathbf{x}$ using only a single latent subspace at a time. First, we find that DIVA is able to reconstruct the original cell images using all three subspaces (Figure 3, second row). Second, we find that the three latent subspaces are indeed disentangled: $\mathbf{z}_d$ is containing the color of the cell (Figure 3, third row), $\mathbf{z}_x$ the shape of the cell (Figure 3, fourth row) and $\mathbf{z}_y$ the location of the parasite (Figure 3, fifth row). The holes in the reconstructions using only $\mathbf{z}_x$ indicate that there is no probability mass in $\mathbf{z}_d$ and $\mathbf{z}_y$ at 0, similar to Figure 2. From the reconstructions in Figure 3 we conclude that DIVA is able to learn disentangled representations that match the ground truth factors of interest, here, the appearance of the cell and the presence of the parasite. In addition to these qualitative results, we show that a classifier for $y$ trained on $\mathbf{z}_d$ or $\mathbf{z}_x$ performs worse than a classifier that would always predict the majority class, the results can be found in the Appendix.

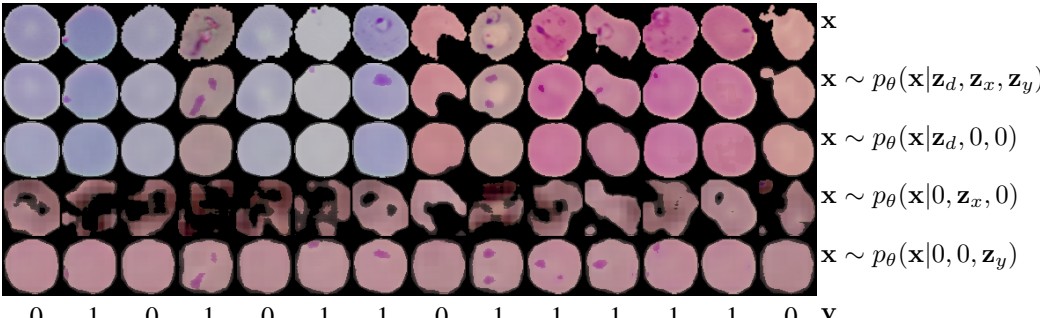

Figure 3: Reconstructions of $\mathbf{x}$ using all three latent subspaces as well as reconstructions of $\mathbf{x}$ using only a single latent subspace at a time.

### 4.2.2 SUPERVISED CASE

To further evaluate domain generalization abilities, models are trained on nine domains (patient IDs) and tested on the remaining 10th domain. We choose ROC AUC on the hold out test domain as the evaluation metric, since the two classes are highly imbalanced. All experiments are repeated five times.

Table 3: Results of the supervised experiments for the first part of domains. We report the average and standard error of ROC AUC.

| Model | C116P77 | C132P93 | C137P98 | C180P141 | C182P143 | C184P145 |
|---|---|---|---|---|---|---|
| Baseline | $90.6 \pm 0.7$ | $97.8 \pm 0.5$ | $98.9 \pm 0.2$ | $98.5 \pm 0.2$ | $96.7 \pm 0.4$ | $98.1 \pm 0.2$ |
| DA | $90.6 \pm 1.7$ | $\mathbf{98.3 \pm 0.4}$ | $99.0 \pm 0.1$ | $98.8 \pm 0.1$ | $96.9 \pm 0.4$ | $97.1 \pm 0.8$ |
| DIVA | $\mathbf{93.3 \pm 0.4}$ | $\mathbf{98.4 \pm 0.3}$ | $99.0 \pm 0.1$ | $\mathbf{99.0 \pm 0.1}$ | $96.5 \pm 0.3$ | $\mathbf{98.5 \pm 0.3}$ |

Table 4: Results of the supervised experiments for the second part of domains. As well as the average across all domains. We report the average and standard error of ROC AUC.

| Model | C39P4 | C59P20 | C68P29 | C99P60 | Average |
|---|---|---|---|---|---|
| Baseline | $97.1 \pm 0.4$ | $82.8 \pm 2.8$ | $95.3 \pm 0.6$ | $96.2 \pm 0.1$ | $95.2 \pm 1.6$ |
| DA | $97.4 \pm 0.3$ | $83.2 \pm 3.3$ | $\mathbf{96.3 \pm 0.1}$ | $96.1 \pm 0.3$ | $95.4 \pm 1.6$ |
| DIVA | $\mathbf{97.8 \pm 0.2}$ | $82.1 \pm 3.0$ | $\mathbf{96.3 \pm 0.2}$ | $\mathbf{96.6 \pm 0.3}$ | $95.8 \pm 1.6$ |

We compare DIVA with a ResNet-like (He et al., 2015) baseline and DA. During inference all three models have the same architecture, seven ResNet blocks followed by two linear layers. Detailed information about hyperparameters, architecture and training schedule can be found in the Appendix. In Table 3 and 4 we find that the results are not equally distributed across all test domains. In five cases DIVA is able to significantly improve upon the baseline model and DA. Upon visual inspection we find that cells from domain C116P77 and domain C59P20 are stained pink, similar to the stain of the parasite, see Appendix. In case of C116P77 DIVA achieves the highest ROC AUC of all three models. In case of domain C59P20, all three methods have difficulties to detect the parasite which leads to the lowest ROC AUC among all domains. We believe that the difficulties arise the poor contrast between cell and parasite. Last, DIVA is able to improve on average when compared to the baseline model and DA, although the improvements are within the standard error.

### 4.2.3 SEMI-SUPERVISED CASE

As described in Section 4.1.3 we are interested in learning from domains with no class labels, since such an approach can drastically lower the amount of labeled data needed to learn a domain invariant representation, i.e., a model that generalizes well across patients. For the semi-supervised experiments we choose domain C116P77 to be the test domain since its cells show a unique dark pink stain. Furthermore, unlabeled data from domain C59P20 is used since it is visually the closest to domain C116P77, see Appendix. The evaluation metric on the hold out test domain is ROC AUC again. In Table 5 we compare the baseline model, DA and DIVA trained with labeled data from domain C59P20, unlabeled data from domain C59P20 and no data from domain C59P20.

We argue that the improvement of DIVA over DA arises from the way the additional unlabeled data is utilized. In case of DA the unlabeled data $(d, \mathbf{x})$ is only used to train the domain classifier and the feature extractor in an adversarial manner. In Section 2.2 we show that due to DIVA's generative nature $q_{\phi_y}(\mathbf{z}_y|\mathbf{x})$, $p_{\theta_y}(\mathbf{z}_y|y)$ can be updated using $q_{\omega_y}(y|\mathbf{z}_y)$ to marginalize over $y$ for an unlabeled sample $\mathbf{x}$. In addition, the unlabeled data $(d, \mathbf{x})$ is used to update $q_{\phi_d}(\mathbf{z}_d|\mathbf{x})$, $p_{\theta_d}(\mathbf{z}_d|d)$, $q_{\omega_d}(d|\mathbf{z}_d)$, $q_{\phi_x}(\mathbf{z}_x|\mathbf{x})$ and $p_\theta(\mathbf{x}|\mathbf{z}_d, \mathbf{z}_x, \mathbf{z}_y)$ in the same way as in the supervised case.

Table 5: Results of the semi-supervised experiments for domain C116P77. Comparison of baseline method, DA and DIVA trained with labeled data from domain C59P20, unlabeled data from domain C59P20 and no data from domain C59P20. We report the average and standard error of ROC AUC.

| Training data | Baseline | DA | DIVA |
|---|---|---|---|
| Labeled data from C59P20 | $90.6 \pm 0.7$ | $90.6 \pm 1.7$ | $\mathbf{93.3 \pm 0.4}$ |
| Unlabeled data from C59P20 | - | $72.05 \pm 2.2$ | $\mathbf{79.4 \pm 2.8}$ |
| No data from C59P20 | $70.0 \pm 2.6$ | $69.2 \pm 1.9$ | $71.9 \pm 2.7$ |

## 5   CONCLUSION

We have proposed DIVA as a generative model with three latent subspaces. We evaluated DIVA on Rotated MNIST and a Malaria Cell Images dataset. In both cases DIVA is able to learn disentangled representations that match the ground truth factors of interest, represented by the class $y$ and the domain $d$. By learning representations $\mathbf{z}_y$ that are invariant with respect to the domain $d$ DIVA is able to improve upon other methods on both datasets. Furthermore, we show that we can boost DIVA's performance by incorporating unlabeled samples, even from entirely new domains for which no labeled examples are available. This property is highly desirable in fields like medical imaging where the labeling process is very time consuming and costly.

In all of our experiments it appears that there is a key difference between interpolation and extrapolation, a distinction currently not made by the domain generalization community: If we assume that the domains lie in intervals like [0°,15°, 30°] or ['red', 'orange', 'yellow'] then the performance for the domains in the center of the interval, e.g., 15° and 'orange', seems to be better than for the domains on the ends of the interval. We argue that DIVA can make use of unlabeled data from a domain that is close to the test domain to improve its extrapolation performance, as seen in Section 4.1.3 and 4.2.3.

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

## APPENDIX

### 5.1  ROTATED MNIST

#### 5.1.1  TRAINING PROCEDURE

All DIVA models are trained for 500 epochs. The training is terminated if the training loss for $y$ has not improved for 100 epochs. As proposed in Burgess et al. (2018), we linearly increase $\beta$ from 0.0 to 1.0 during the first 100 epochs of training. We set $\alpha_d = 2000$. As seen in (Maaløe et al., 2019), we adjust $\alpha_y$ according to the ratio of labeled (N) and unlabeled data (M),

$$\alpha_y = \gamma \frac{M + N}{N}, \tag{5}$$

where we set $\gamma = 3500$. Last, $\mathbf{z}_d$, $\mathbf{z}_x$ and $\mathbf{z}_y$ each have 64 latent dimensions. All hyperparameters were determined by training DIVA on $\mathcal{M}_{0°}$, $\mathcal{M}_{15°}$, $\mathcal{M}_{30°}$, $\mathcal{M}_{45°}$ and testing on $\mathcal{M}_{60°}$. We searched over the following parameters: $\alpha_d$, $\alpha_d \in \{1500, 2000, 2500, 3000, 3500, 4000\}$; $\dim(\mathbf{z}_d) = \dim(\mathbf{z}_x)$ = $\dim(\mathbf{z}_y)$ and $\dim(\mathbf{z}_x) \in \{16, 32, 64\}$; $\beta_{max} \in \{1, 5, 10\}$.

All models were trained using ADAM (Kingma & Ba, 2014) (with default settings), a pixel-wise cross entropy loss and a batch size of 100.

#### 5.1.2  ARCHITECTURES

To enable a fair experiment, the encoder $q_{\phi_y}(\mathbf{z}_y|\mathbf{x})$ and auxiliary classifier $q_{\omega_y}(y|\mathbf{z}_y)$ form a DNN with the same number of layers and weights as described in Wang et al. (2019).

Table 6: Architecture for $p_\theta(\mathbf{x}|\mathbf{z}_d, \mathbf{z}_x, \mathbf{z}_y)$. The parameter for Linear is output features. The parameters for ConvTranspose2d are output channels and kernel size. The parameter for Upsample is the upsampling factor. The parameters for Conv2d are output channels and kernel size.

| block | details |
|-------|---------|
| 1 | Linear(1024), BatchNorm1d, ReLU |
| 2 | Upsample(2) |
| 3 | ConvTranspose2d(128, 5), BatchNorm2d, ReLU |
| 4 | Upsample(2) |
| 5 | ConvTranspose2d(256, 5), BatchNorm2d, ReLU |
| 6 | Conv2d(256, 1) |

Table 7: Architecture for $p_{\theta_d}(\mathbf{z}_d|d)$ and $p_{\theta_y}(\mathbf{z}_y|y)$. Each network has two heads one for the mean and one for the scale. The parameter for Linear is output features.

| block | details |
|-------|---------|
| 1 | Linear(64), BatchNorm1d, ReLU |
| 2.1 | Linear(64) |
| 2.2 | Linear(64), Softplus |

Table 8: Architecture for $q_{\phi_d}(\mathbf{z}_d|\mathbf{x})$, $q_{\phi_x}(\mathbf{z}_x|\mathbf{x})$ and $q_{\phi_y}(\mathbf{z}_y|\mathbf{x})$. Each network has two heads one for the mean one and for the scale. The parameters for Conv2d are output channels and kernel size. The parameters for MaxPool2d are kernel size and stride. The parameter for Linear is output features.

| block | details |
|---|---|
| 1 | Conv2d(32, 5), BatchNorm2d, ReLU |
| 2 | MaxPool2d(2, 2) |
| 3 | Conv2d(64, 5), BatchNorm2d, ReLU |
| 4 | MaxPool2d(2, 2) |
| 5.1 | Linear(64) |
| 5.2 | Linear(64), Softplus |

Table 9: Architecture for $q_{\omega_d}(d|\mathbf{z}_d)$ and $q_{\omega_y}(y|\mathbf{z}_y)$. The parameter for Linear is output features.

| block | details |
|---|---|
| 1 | ReLU, Linear(5 for $q_{\omega_d}(d|\mathbf{z}_d)$/10 for $q_{\omega_y}(y|\mathbf{z}_y)$), Softmax |

### 5.1.3 SAMPLES

We present samples from DIVA by sampling $\mathbf{z}_d$, $\mathbf{z}_x$ and $\mathbf{z}_y$ from their priors and then decoding them. Generated examples on the Rotated MNIST data are given in the Figure 4. DIVA allows to generate images that are almost indistinguishable from real datapoints.

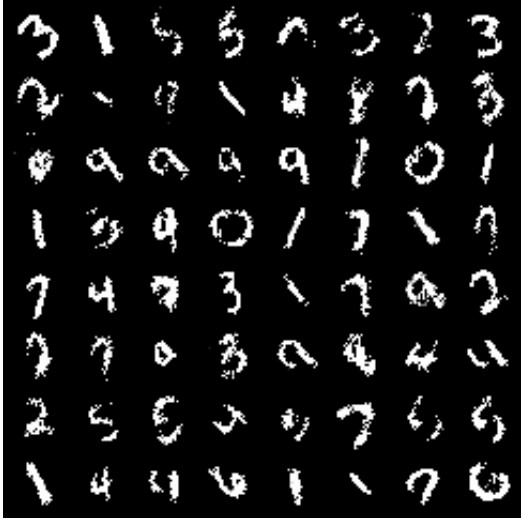

Figure 4: Samples from DIVA trained on Rotated MNIST.

### 5.1.4 CONDITIONAL GENERATION

Yet another way to gain insight into the disentanglement abilities of DIVA is conditional generation. We first train DIVA with $\beta = 10$ using $\mathcal{M}_{0°}$, $\mathcal{M}_{15°}$, $\mathcal{M}_{30°}$, $\mathcal{M}_{45°}$ and $\mathcal{M}_{60°}$ as training domains. After training we perform two experiments. In the first one we are fixing the class and varying the domain. In the second experiment we are fixing the domain and varying the class.

**Change of class** The first row of Figure 5 (left) shows the input images $x$ for DIVA. First, we generate embeddings $\mathbf{z}_d$, $\mathbf{z}_x$ and $\mathbf{z}_y$ for each $\mathbf{x}$ using $q_{\phi_d}(\mathbf{z}_d|\mathbf{x})$, $q_{\phi_x}(\mathbf{z}_x|\mathbf{x})$ and $q_{\phi_y}(\mathbf{z}_y|\mathbf{x})$. Second, we replace $\mathbf{z}_y$ with a sample $\mathbf{z}_y'$ from the conditional prior $p_{\theta_y}(\mathbf{z}_y|y)$. Last, we generate new images from $\mathbf{z}_d$, $\mathbf{z}_x$ and $\mathbf{z}_y'$ using the trained encoder $p_\theta(\mathbf{x}|\mathbf{z}_d, \mathbf{z}_x, \mathbf{z}_y)$. In Figure 5 (left) rows 2 to 11

correspond to the classes '0' to '9'. We observe that the rotation angle (encoded in $\mathbf{z}_d$) and the line thickness (encoded in $\mathbf{z}_x$) are well preserved, while the class of the image is changing as intended.

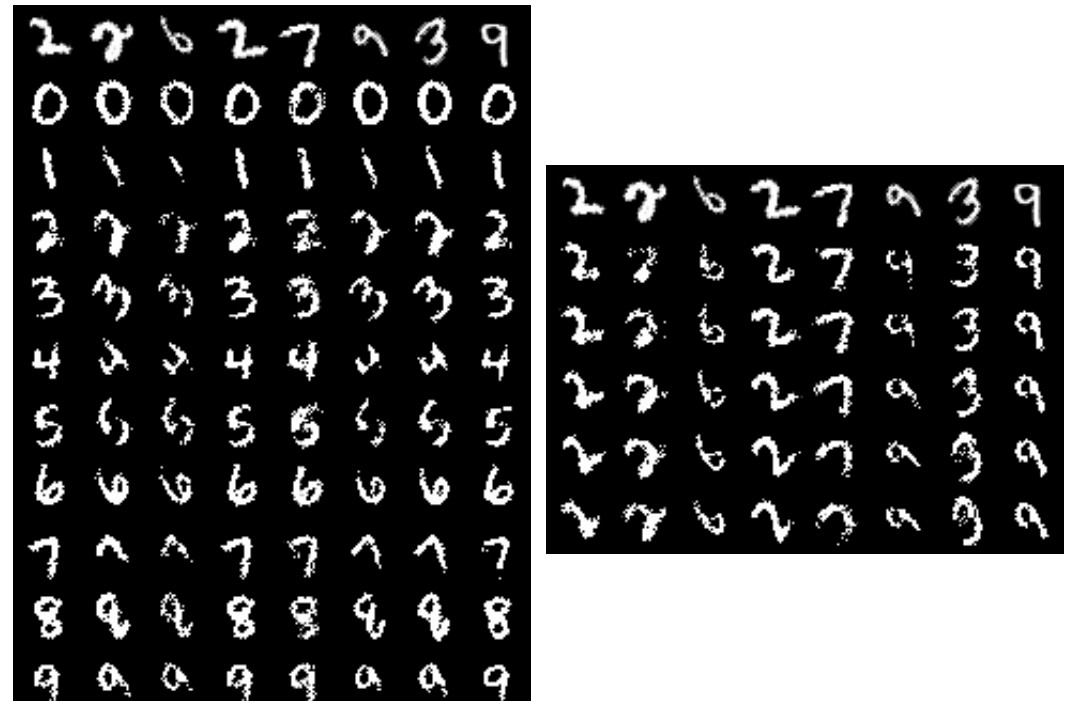

Figure 5: Reconstructions. Left: First row is input, row 2 to 11 correspond to labels '0' to '9'. Right: First row is input, row 2 to 6 correspond to domains 0, 15, 30, 45, 60.

**Change of domain** We repeat the experiment from above but this time we keep $\mathbf{z}_x$ and $\mathbf{z}_y$ fixed while changing the domain. After generating embeddings $\mathbf{z}_d$, $\mathbf{z}_x$ and $\mathbf{z}_y$ for each $\mathbf{x}$ in the first row of Figure 5 (right), we replace $\mathbf{z}_d$ with a sample $\mathbf{z}_d'$ from the conditional prior $p_{\theta_d}(\mathbf{z}_d|d)$. Finally, we generate new images from $\mathbf{z}_d'$, $\mathbf{z}_x$ and $\mathbf{z}_y$ using the trained encoder $p_\theta(\mathbf{x}|\mathbf{z}_d, \mathbf{z}_x, \mathbf{z}_y)$. In Figure 5 (right) rows 2 to 6 correspond to the domains $\mathcal{M}_{0^\circ}$ to $\mathcal{M}_{60^\circ}$. Again, DIVA shows the desired behaviour: While the rotation angle is changing the class and style of the original image is maintained.

### 5.1.5 QUALITATIVE DISENTANGLEMENT: TEST DOMAIN

In this section, we visualize the $\mathbf{z}_d$ and $\mathbf{z}_y$ for data points $x$ from the test domain $\mathcal{M}_{75^\circ}$ for the model trained in Section 4.1.1. Figure 7 shows 1000 embeddings $\mathbf{z}_y$ encoded by $q_{\phi_y}(\mathbf{z}_y|\mathbf{x})$. Figure 6 shows 1000 embeddings $\mathbf{z}_d$ encoded by $q_{\phi_d}(\mathbf{z}_d|\mathbf{x})$.

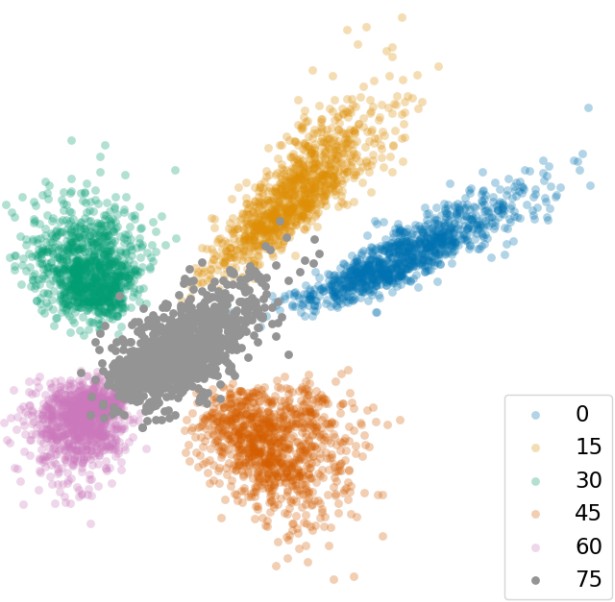

Figure 6: 1000 two-dimensional embeddings $\mathbf{z}_d$ encoded by $q_{\phi_d}(\mathbf{z}_d|\mathbf{x})$ for $\mathbf{x}$ from the test domain $\mathcal{M}_{75°}$. The color of each point indicates the associated domain.

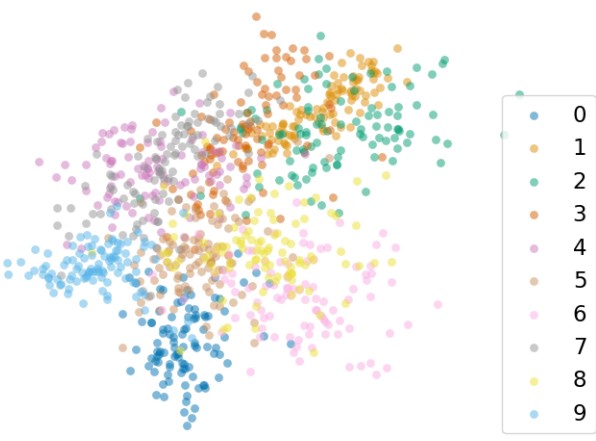

Figure 7: 1000 two-dimensional embeddings $\mathbf{z}_y$ encoded by $q_{\phi_y}(\mathbf{z}_y|\mathbf{x})$ for $\mathbf{x}$ from the test domain $\mathcal{M}_{75°}$. The color of each point indicates the associated class.

### 5.1.6 ABLATION STUDY: PARTITIONED LATENT SPACE

We compare DIVA to a VAE with a single latent space, a standard Gaussian prior and two auxillary tasks. The resulting graphical model is shown in Figure 8. The results in Table 10 clearly show the benefits of having a partitioned latent space $\mathbf{z}$.

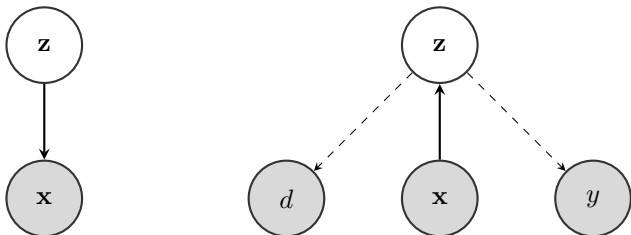

Figure 8: Left: Generative model. According to the graphical model we obtain $p(\mathbf{x}, \mathbf{z}) = p_\theta(\mathbf{x}|\mathbf{z})p(\mathbf{z})$. Right: Inference model. We propose $q_\phi(\mathbf{z}|\mathbf{x})$ as the variational posterior. Dashed arrows represent the two auxiliary classifiers $q_{\omega_d}(d|\mathbf{z})$ and $q_{\omega_y}(y|\mathbf{z})$.

The objective is given by,

$$\mathcal{F}_{\text{VAE}}(d, \mathbf{x}, y) := \mathbb{E}_{q_\phi(\mathbf{z}|\mathbf{x})}\left[\log p_\theta(\mathbf{x}|\mathbf{z})\right] - \beta KL\left(q_\phi(\mathbf{z}|\mathbf{x})||p(\mathbf{z})\right)$$
$$+ \alpha_d \mathbb{E}_{q_\phi(\mathbf{z}|\mathbf{x})}\left[\log q_{\omega_d}(d|\mathbf{z})\right] + \alpha_y \mathbb{E}_{q_\phi(\mathbf{z}|\mathbf{x})}\left[\log q_{\omega_y}(y|\mathbf{z})\right]. \quad (6)$$

Table 10: Comparison of DIVA with a VAE with a single latent space, a standard Gaussian prior and two auxillary tasks on Rotated MNIST. We report the average and standard error of the classification accuracy.

| Test | VAE | DIVA |
|------|-----|------|
| $\mathcal{M}_{0°}$ | $88.4 \pm 0.5$ | $\mathbf{93.5 \pm 0.3}$ |
| $\mathcal{M}_{15°}$ | $98.3 \pm 0.1$ | $\mathbf{99.3 \pm 0.1}$ |
| $\mathcal{M}_{30°}$ | $97.4 \pm 0.2$ | $\mathbf{99.1 \pm 0.1}$ |
| $\mathcal{M}_{45°}$ | $97.4 \pm 0.4$ | $\mathbf{99.2 \pm 0.1}$ |
| $\mathcal{M}_{60°}$ | $97.9 \pm 0.2$ | $\mathbf{99.3 \pm 0.1}$ |
| $\mathcal{M}_{75°}$ | $84.0 \pm 0.3$ | $\mathbf{93.0 \pm 0.4}$ |
| Avg | $93.9 \pm 0.1$ | $\mathbf{97.2 \pm 1.3}$ |

### 5.1.7 ABLATION STUDY: DIVA WITHOUT $z_d$ AND $z_x$

We compare DIVA as proposed in Section 2.1 to two ablated versions of DIVA:

1. DIVA without $z_d$: The domain label $d$ is not used during training. Therefore, there exist no latent space $z_d$, no encoder $q_{\phi_d}(z_d|x)$, no prior $p_{\theta_d}(z_d|d)$ and no classifier $q_{\omega_d}(d|z_d)$. The decoder becomes $p_\theta(x|z_x, z_y)$.

2. DIVA without $z_x$: There exist no latent space $z_x$, no encoder $q_{\phi_x}(z_x|x)$ and no prior $p(z_x)$. The decoder becomes $p_\theta(x|z_d, z_y)$.

In Table 11, we compare DIVA and the two ablated versions on the Rotated MNIST dataset. Surprisingly, for Rotated MNIST we could not find a significant difference in performance between DIVA and DIVA without $z_d$, as seen in the third column. However, not having $z_d$ drastically reduces the interpretability of our model, since without $z_d$ we cannot find the variations in x that are explained by the domain $d$. E.g. in Appendix 5.1.4, we show that we can generate samples conditioned on the domain label that give us a clear idea of the meaning of $d$. Furthermore, as seen in Figure 3, we see that the patient ID is highly correlated with the color of the stain. While the cell shape in not correlated with $d$ or $y$ and therefore is captured by $z_x$. Without $z_d$ we are unable to gain such (especially from a medical perspective) important insights. In the fourth column, we see that for $\mathcal{M}_{0°}$ and $\mathcal{M}_{75°}$ DIVA with $z_x$ performs significantly better than without. We argue that if $z_x$ does not exist, $z_d$ and $z_y$ will capture the residual variations in x that are not explained by $d$ or $y$. We believe this makes it harder to predict $y$ using $z_y$ and $d$ using $z_d$.

Table 11: Results of ablation study.

| Test | DIVA | DIVA without $z_d$ | DIVA without $z_x$ |
|---|---|---|---|
| $\mathcal{M}_{0°}$ | $93.5 \pm 0.3$ | $93.4 \pm 0.5$ | $92.7 \pm 0.5$ |
| $\mathcal{M}_{15°}$ | $99.3 \pm 0.1$ | $99.3 \pm 0.1$ | $99.4 \pm 0.1$ |
| $\mathcal{M}_{30°}$ | $99.1 \pm 0.1$ | $98.9 \pm 0.1$ | $99.2 \pm 0.1$ |
| $\mathcal{M}_{45°}$ | $99.2 \pm 0.1$ | $99.1 \pm 0.1$ | $99.1 \pm 0.1$ |
| $\mathcal{M}_{60°}$ | $99.3 \pm 0.1$ | $99.1 \pm 0.1$ | $99.4 \pm 0.1$ |
| $\mathcal{M}_{75°}$ | $93.0 \pm 0.4$ | $92.8 \pm 0.4$ | $92.4 \pm 0.4$ |
| Avg | $97.2 \pm 1.3$ | $97.1 \pm 1.3$ | $97.1 \pm 1.5$ |

## 5.2 MALARIA CELL IMAGES

### 5.2.1 EXAMPLE CELLS FROM EACH DOMAIN

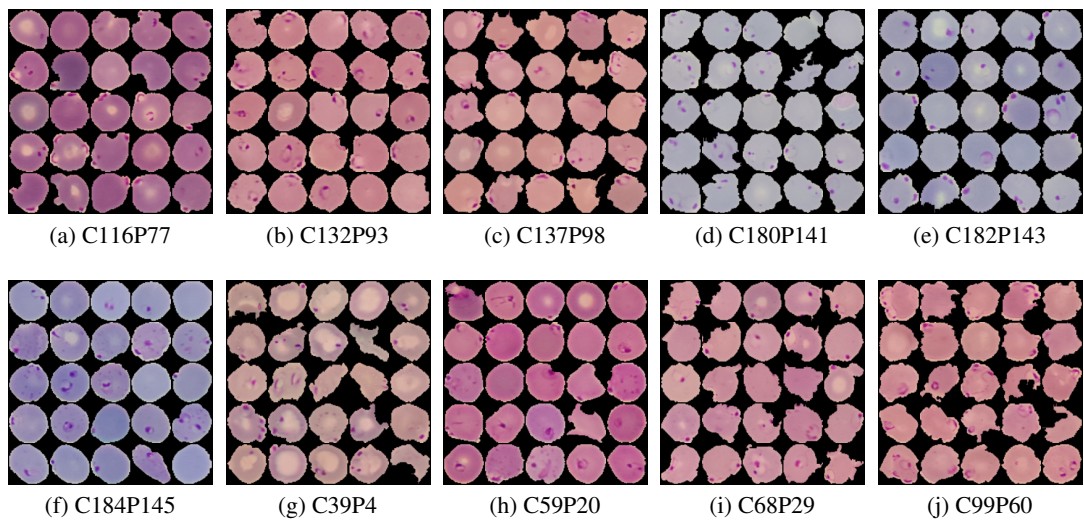

| (a) C116P77 | (b) C132P93 | (c) C137P98 | (d) C180P141 | (e) C182P143 |
|---|---|---|---|---|

| (f) C184P145 | (g) C39P4 | (h) C59P20 | (i) C68P29 | (j) C99P60 |
|---|---|---|---|---|

### 5.2.2 TRAINING PROCEDURE: DIVA

All DIVA models are trained for 500 epochs. The training is terminated if the validation accuracy for $y$ has not improved for 100 epochs. As proposed in Burgess et al. (2018), we linearly increase $\beta$ from 0.0 to 1.0 during the first 100 epochs of training. We set $\alpha_d = 100000$ and $\alpha_y = 75000$. Last, $\mathbf{z}_d$, $\mathbf{z}_x$ and $\mathbf{z}_y$ each have 64 latent dimensions. We searched over the following parameters: $\alpha_d$, $\alpha_d \in \{25000, 50000, 75000, 100000\}$; $\dim(\mathbf{z}_d) = \dim(\mathbf{z}_x) = \dim(\mathbf{z}_y)$, $\dim(\mathbf{z}_x) \in \{32, 64\}$; $\beta_{max} \in \{1, 5, 10\}$. All hyperparameters were determined using a validation set that consists of 20 % of the training set. All models were trained using ADAM (Kingma & Ba, 2014) (with default settings), a mixture of discretized logistics (Salimans et al., 2017) loss and a batch size of 100. In case of the semi-supervised experiment in Section 4.2.3 we adapt $\alpha_d$ and $\alpha_y$ according to Equation 5.

### 5.2.3 TRAINING PROCEDURE: BASELINE AND DA

In case of the supervised experiments in Section 5.2.5 all models are trained for 500 epochs. The training is terminated if the validation accuracy for $y$ has not improved for 100 epochs. In case of the semi-supervised experiments in Section 2.2 the amount of epochs is adjusted to match the number of parameter updates of DIVA. For DA we follow the same training procedure as described in Ganin et al. (2015). In the supervised case, first, a labeled batch randomly sampled from the training distributions is used to update the class classifier, domain classifier and the feature extractor in an adversarial fashion. Second, a second batch randomly sampled from the training distributions is used to update only the domain classifier and the feature extractor in an adversarial fashion. In the semi-supervised case samples from the unsupervised domains form the second batch together

with samples from the supervised domains. We use the same domain adaptation parameter $\lambda$ schedule as described in Ganin et al. (2015). Determined by hyperparameter search we find that DA performs better when $\lambda \cdot \epsilon$ is used. Here, $\epsilon = 0.001$. We searched over the following values of $\epsilon \in \{0.1, 0.05, 0.01, 0.005, 0.001, 0.0005, 0.0001\}$. In case of the semi-supervised experiment in Section 4.2.3 $\epsilon = 0.01$ was determined by hyperparameter search.

### 5.2.4 ARCHITECTURE

In the following we will describe the architecture of DIVA in detail. Note that the architecture for the baseline model is the same as $q_{\phi_y}(\mathbf{z}_y|\mathbf{x})$ (we only use the mean of $\mathbf{z}_y$) followed by $q_{\omega_y}(y|\mathbf{z}_y)$ where $\mathbf{z}_y$ has 1024 dimensions. DA is using $q_{\phi_y}(\mathbf{z}_y|\mathbf{x})$ without the linear layer as a feature extractor. The class classifier and the domain classifier consist of two linear layers. The feature extractor for all models consist of seven ResNet blocks (He et al., 2015). During training batch norm Ioffe & Szegedy (2015) is used for all layers.

Table 12: Architecture for $p_\theta(\mathbf{x}|\mathbf{z}_d, \mathbf{z}_x, \mathbf{z}_y)$. The parameter for Linear is output features. The parameters for ResidualConvTranspose2d are output channels and kernel size. The parameters for Conv2d are output channels and kernel size.

| block | details |
|-------|---------|
| 1 | Linear(1024), BatchNorm1d, LeakyReLU |
| 2 | ResidualConvTranspose2d(64, 3), LeakyReLU |
| 3 | ResidualConvTranspose2d(64, 3), LeakyReLU |
| 4 | ResidualConvTranspose2d(64, 3), LeakyReLU |
| 5 | ResidualConvTranspose2d(32, 3), LeakyReLU |
| 6 | ResidualConvTranspose2d(32, 3), LeakyReLU |
| 7 | ResidualConvTranspose2d(32, 3), LeakyReLU |
| 8 | ResidualConvTranspose2d(32, 3), LeakyReLU |
| 9 | ResidualConvTranspose2d(32, 3), LeakyReLU |
| 10 | Conv2d(100, 3) |
| 11 | Conv2d(100, 1) |

Table 13: Architecture for $p_{\theta_d}(\mathbf{z}_d|d)$ and $p_{\theta_y}(\mathbf{z}_y|y)$. Each network has two heads one for the mean and one for the scale. The parameter for Linear is output features.

| block | details |
|-------|---------|
| 1 | Linear(64), BatchNorm1d, LeakyReLU |
| 2.1 | Linear(64) |
| 2.2 | Linear(64), Softplus |

Table 14: Architecture for $q_{\phi_d}(\mathbf{z}_d|\mathbf{x})$, $q_{\phi_x}(\mathbf{z}_x|\mathbf{x})$ and $q_{\phi_y}(\mathbf{z}_y|\mathbf{x})$. Each network has two heads one for the mean one and for the scale. The parameters for Conv2d are output channels and kernel size. The parameters for ResidualConv2d are output channels and kernel size. The parameter for Linear is output features.

| block | details |
|---|---|
| 1 | Conv2d(32, 3), BatchNorm2d, LeakyReLU |
| 2 | ResidualConv2d(32), LeakyReLU |
| 3 | ResidualConv2d(32), LeakyReLU |
| 4 | ResidualConv2d(64, 3), LeakyReLU |
| 5 | ResidualConv2d(64, 3), LeakyReLU |
| 6 | ResidualConv2d(64, 3), LeakyReLU |
| 7 | ResidualConv2d(64, 3), LeakyReLU |
| 8 | ResidualConv2d(64, 3), LeakyReLU |
| 9.1 | Linear(64) |
| 9.2 | Linear(64), Softplus |

Table 15: Architecture for $q_{\omega_d}(d|\mathbf{z}_d)$ and $q_{\omega_y}(y|\mathbf{z}_y)$. The parameter for Linear is output features.

| block | details |
|---|---|
| 1 | LeakyReLU, Linear(9 for $q_{\omega_d}(d|\mathbf{z}_d)$/2 for $q_{\omega_y}(y|\mathbf{z}_y)$), Softmax |

### 5.2.5 PREDICTING $y$ USING EITHER $\mathbf{z}_d$, $\mathbf{z}_x$ OR $\mathbf{z}_y$

We test how predictive $\mathbf{z}_d$, $\mathbf{z}_x$ and $\mathbf{z}_y$ are for the class $y$ on the Malaria Cell Images dataset. First, we use the trained DIVA models from to create embeddings $\mathbf{z}_d$, $\mathbf{z}_x$ and $\mathbf{z}_y$ for every $\mathbf{x}$ in the training domain and hold out test domain. Second, we train a 2-layer MLP on the embeddings $\mathbf{z}_d$, $\mathbf{z}_x$ and $\mathbf{z}_y$ from the training domains. We train the MLP for 100 epochs using ADAM (Kingma & Ba, 2014). After training we test the MLP embeddings $\mathbf{z}_d$, $\mathbf{z}_x$ and $\mathbf{z}_y$ from the test domain. In Table 16 we clearly see that $\mathbf{z}_y$ captures all relevant information in order to predict $y$, while the MLPs trained using $\mathbf{z}_d$ and $\mathbf{z}_x$ perform worse than a classifier that would always pick the majority class.

Table 16: Prediction of $y$ using a 2 layer MLP trained using $\mathbf{z}_d$, $\mathbf{z}_x$ and $\mathbf{z}_y$. We report the mean and standard error of the classification accuracy on the hold out test domain.

| test domain | $\mathbf{z}_d$ | $\mathbf{z}_x$ | $\mathbf{z}_y$ | majority class |
|---|---|---|---|---|
| 0 | $84.6 \pm 1.0$ | $85.0 \pm 0.2$ | $\mathbf{87.9 \pm 0.9}$ | 0.86 |
| 1 | $89.5 \pm 0.4$ | $88.2 \pm 0.5$ | $\mathbf{96.8 \pm 0.1}$ | 0.9 |
| 2 | $68.2 \pm 3.5$ | $80.0 \pm 1.6$ | $\mathbf{96.9 \pm 0.5}$ | 0.81 |
| 3 | $87.0 \pm 0.3$ | $75.2 \pm 2.9$ | $\mathbf{95.5 \pm 0.2}$ | 0.88 |
| 4 | $89.1 \pm 0.3$ | $82.7 \pm 2.4$ | $\mathbf{92.5 \pm 0.4}$ | 0.90 |
| 5 | $88.3 \pm 0.2$ | $87.7 \pm 0.2$ | $\mathbf{90.6 \pm 0.5}$ | 0.88 |
| 6 | $82.6 \pm 3.7$ | $56.3 \pm 5.1$ | $\mathbf{91.1 \pm 0.1}$ | 0.90 |
| 7 | $88.3 \pm 0.1$ | $88.3 \pm 0.1$ | $\mathbf{90.8 \pm 0.8}$ | 0.88 |
| 8 | $89.5 \pm 0.3$ | $85.3 \pm 1.7$ | $\mathbf{93.5 \pm 0.4}$ | 0.90 |
| 9 | $89.1 \pm 0.2$ | $86.9 \pm 1.5$ | $\mathbf{94.0 \pm 0.3}$ | 0.89 |

