# OpenReview forum: "DIVA: Domain Invariant Variational Autoencoder"
_ICLR.cc/2020/Conference — Reject_

### Official Review · AnonReviewer1 · 2019-10-16
**Official Blind Review #1**

**Rating:** 6

**Review:**

The paper introduces a VAE that can be used in problems in which domain information is available at training time to increase the classification performances on unseen domains. The model can also be used in a semi-supervised setting if unlabelled data is available (even from unseen domains).

As the authors state, this is a quite common scenario in many interesting applications such as medical imaging. As such, I found the paper interesting to read (the paper is also written quite well).

To allow the classification of data from any domain, in the inference network the labels d are not used to infer the latent states, but only as an auxiliary loss that forces z_d to capture domain-specific information. However, this makes me wonder if z_d is needed at all? I wouldn't be surprised if the model performed equally well if domain information d was not passed to the model even during training, in which case z_x would also capture domain-specific information. To understand if this is the case, it would be very helpful to add a baseline model in which you use a version of DIVA in which z_d and d are not present both in the generative model and in the inference network.

Since from the technical point of view the novelty of the model is limited, I would have liked to see a stronger experimental section to show the real-life applicability of the model:
- The MNIST experiments are visually helpful to understand the disentanglement in the model. However you solve a quite simple task. What are the performances of a baseline classifier on this?
- The malaria cell experiment is definitely more realistic and therefore interesting, and the results are quite convincing (despite being on quite low-resolution images). Since I consider the MNIST experiment a "toy task", I think the paper would greatly improve if a new real-world experiment was performed (e.g. other medical imaging datasets).

Overall I liked the paper and I think it is relevant for the ICLR community, therefore I am voting towards acceptance. However, a more convincing experimental section is needed for me to increase the score.


Small comments:
- you should make clearer from the abstract that the goal of this paper is to build a domain-invariant classifier (as opposed to solving more VAE-like tasks)
- in the beginning of section 2.1 you write twice p(x|z_d, z_x, z_d), without z_y


**Experience Assessment:**

I have published in this field for several years.

**Review Assessment: Checking Correctness Of Derivations And Theory:**

I assessed the sensibility of the derivations and theory.

**Review Assessment: Checking Correctness Of Experiments:**

I carefully checked the experiments.

**Review Assessment: Thoroughness In Paper Reading:**

I read the paper at least twice and used my best judgement in assessing the paper.

---

> ### Author Response · Authors · 2019-11-13
> **Rebuttal**
>
>
> 1. Is z_d needed at all
> - In order to achieve good reconstructions the latent space of DIVA needs to capture variations in x which are domain specific (e.g., color as seen for Malaria Cell Images). Without z_d, either z_x or z_y have to capture these variations. In the worst case scenario, z_y is capturing all domain specific variations in x. This is likely to happen when information in d can be exploited to predict y. As a result, DIVA will fail to generalize to unseen domains.
> - Surprisingly, for Rotated MNIST we could not find a significant difference in performance between DIVA and DIVA without z_d, see Table 11. However, not having z_d drastically reduces the interpretability of our model, since without z_d we cannot find the variations in x that are explained by the domain d. E.g. in appendix 5.1.4, we show that we can generate samples conditioned on the domain label that give us a clear idea of the meaning of d. Furthermore, as seen in Figure 3, we see that the patient ID is highly correlated with the color of the stain. While the cell shape in not correlated with d or y and therefore is captured by z_x. Without z_d we are unable to gain such (especially from a medical perspective) important insights.
>
> 2. Datasets
> - Rotated MNIST is a well established benchmark in the field of domain generalization, used by the majority of papers mentioned in the related work section, e.g., [1, 2, 3, 4].
> - Malaria Cell Images is indeed a very realistic dataset. While the problem of domain shifts is seemingly everywhere in the real world, it is hard to find large open access datasets where the domain labels and class labels are available. We would greatly appreciate any pointers or suggestions.
>
> 3. Small comments
> - We highly appreciate the thorough review! We corrected these typos in the revised version.
>
> [1] - Carlucci et al., 2018 - Agnostic Domain Generalization
> [2] - Ghifary et al., 2015 - Domain Generalization for Object Recognition with Multi-task Autoencoders
> [3] - Motiian et al., 2017 - Unified Deep Supervised Domain Adaptation and Generalization
> [4] - Wang et al., 2019 - Learning Robust Representations by Projecting Superficial Statistics Out

---

> > ### Comment · AnonReviewer1 · 2019-11-13
> > **Thanks**
> >
> > Thanks for your reply and the new experiments in the revised version of the paper. I will leave the score unchanged and will argue for accepting the paper.

---

### Official Review · AnonReviewer3 · 2019-10-23
**Official Blind Review #3**

**Rating:** 3

**Review:**

In this work, the authors propose a domain invariant variational autoencoder for domain generalization problem. Specifically, the data is assumed to be constructed from three independent variables, one for the domain, one for the class and one for the residual variations. The method can be used for both unsupervised and semi-supervised cases. Experimental studies on rotated MNIST dataset and a malaria cell images dataset verify the effectiveness of the proposed method.
The paper is well-written and easy to follow. The proposed generative model is simple and technically sound. However, I have the following concerns.
(1)	It is not clear how the problem setting, i.e., domain generalization, matters in DIVA. In another words, the proposed DIVA is not specific to domain generalization problem, but can be used for domain adaptation, multiple source transfer learning etc. Actually, I find that the authors compare with DA, which is a conventional domain adaptation method, in the experiment. Moreover, the experimental setups in section 4.1.3 is a multi-source transfer setting, and DIVA can be well be applied. In this sense, I am not very convinced on the claim of the contribution that DIVA is proposed for domain generalization. For me, DIVA is a more general method.
(2)	With point 1, more related works on VAE for domain adaptation need to be discussed.
(3)	The idea of constructing data from disentangled latent variables is not new, see a latest work [ref1]. The main difference of DIVA from [ref1] is the residual variation variable. Two baselines are necessary for comparisons: (1) [ref1], and (2) DIVA without the residual variation variable. Actually, the semantic meaning of z_x is not well discussed. Even in the experimental studies figures 2 and 3, it is hard to tell what Z_x actually represents.
(4)	Regarding the 4.1.2, why DA is selected as a baseline? How does DA deal with the multiple domains? Can any other domain adaptation methods, e.g., [ref2], or multiple source transfer methods, e.g. [ref3], be compared?
(5)	In the right side of Table 1, the improvements seem to be very marginal considering the variance. This makes the ability of DIVA to use unlabelled data less convincing.
(6)	Regarding 4.1.3, it seems that the domain similarity plays an important role in the performance, comparing the results of M_{30} with M_{60}. Without the labelled M_{60}, which is very similar to the target M_{75}, the performance degenerates dramatically. The current DIVA treats all the domains equally, is it possible to have a weighted form of DIVA that distinguishes the contributions of different domains?
(7)	What is the task of the malaria cell images experiments? Is it to classify the parasitized and uninfected cells? For a given patient, it makes more sense that all the cells belongs to one category, either infected or healthy. How is the class distribution for a person (in this case a domain)? Is it very unbalanced?
(8)	For figure 3, It is hard to judge the cell images parasitized or uninfected without domain knowledge, can you give the label for each image? Again, the semantic meaning of Z-x is hard to tell. I am not convinced by the shape of the cell for Z-x.
(9)	For 4.2.2, why and how DA is compared? As far as I know, it is for unsupervised domain adaptation. Moreover, the improvements are quite marginal.
Some minor comments:
(1)	Page1, first para, 3rd line, “present” -> “presented”.
(2)	Page2, first para, 2nd line, “Y” - > “Y denotes”
(3)	Some references lack of page information
The paper should be self-contained. I would suggest the authors move some paragraphs in appendix to the paper, for instance, 5.1.1, 5.2.2, and 5.2.3.
Overall, the paper is presented with extensive empirical evaluations, but less theoretical justification. The significance of the paper is moderate as the key idea of learning disentangled latent variables has been studied, and the paper lacks of evidence to show the pure benefits of introducing Z_x as well as the comparison with the related work [ref1].
[ref1] Learning Disentangled Semantic Representation for Domain Adaptation
[ref2] Conditional Adversarial Domain Adaptation
[ref3] Multiple Source Domain Adaptation with Adversarial Learning


**Experience Assessment:**

I have published in this field for several years.

**Review Assessment: Checking Correctness Of Derivations And Theory:**

I carefully checked the derivations and theory.

**Review Assessment: Checking Correctness Of Experiments:**

I assessed the sensibility of the experiments.

**Review Assessment: Thoroughness In Paper Reading:**

I read the paper at least twice and used my best judgement in assessing the paper.

---

> ### Author Response · Authors · 2019-11-13
> **Rebuttal**
>
>
> 1. DIVA for domain generalization
> - We developed DIVA with the clear goal to solve domain generalization problems.  Therefore, domain generalization is the scope of our paper.
> - While developing DIVA we realized that domain generalization is closely related to a number of different research topics, e.g., transfer learning, meta-learning and fairness. In contrast to a transfer learning setting, we do not fit DIVA to the test domain in 4.1.3 or 4.2.3. While it would be interesting to investigate the connections of domain generalizations and other research topics, it is outside of the scope of the paper.
>
> 2. Domain adversarial training for domain generalization
> - While domain adversarial training was initially developed for domain adaptation by now it is widely adopted for domain generalization. It can be found as a baseline in [Wang et al., 2019]. In addition, there are numerous medical imaging paper that are using domain adversarial training in a domain generalization setting, e.g., [4].
>
> 3. More related works on VAE for domain adaptation
> - Thank you very much for the additional references. We tried to be extremely careful during our literature review but we missed the references mentioned by you. We added them to the related work section.
>
> 4. Comparison to Learning Disentangled Semantic Representation for Domain Adaptation
> - We started to work on DIVA in late 2018. Therefore, the work of [1] is concurrent to DIVA.
> - As it can be seen in Figure 2 of [1] they indeed started from basically the same generative model. But there are some core differences. While [1] use gradient reversal layers, we directly parameterize the generative model. As a result, the priors in DIVA are conditional which is necessary for guaranteed disentanglement as recent research has shown [2]. Furthermore [1] apply their model only to domain adaptation. Whereas we are interested in domain generalization. Last, the work of [1] shows the effectiveness of a partitioned latent space as used in DIVA.
>
> 5. Is z_x needed at all
> - From a generative point of view we believe that not all variations in x can be explained by the domain label d and the class label y. We believe that there exist additional variations, e.g., line thickness or cell shape that are independent of the domain label d and the class label y. Therefore we introduce z_x in order to capture these residual variations. If z_x does not exist, z_d and z_y will capture these additional variations in x. We believe this makes it harder to predict y using z_y and d using z_d.
> - We run DIVA without z_x as suggested. The results can be found in Table 11 in the Appendix. Here we clearly see that the results for DIVA without z_x are significantly worse for the test domains M_0 and M_75.
>
> 6. How does Domain adversarial training work for multiple domains
> - A detailed description of the training procedure is given in 5.2.3
> - We tried different setups, the one that worked best was feeding two batches at the same time: one batch with class labels y and domain labels d, a second batch with only domain labels d. This is close to the original update schedule in [3].
>
> 7. Semi-supervised results in Table 1
> - While on average the improvements are small, there are test domains for which the improvements are significant, especially when tested on M_75.
>
> 8. A weighted form of DIVA that distinguishes the contributions of different domains?
> - This is a very interesting idea! Right now we do not see a simple way of detecting how close two domains are to each other. We could explore this in future work.
>
> 9. Malaria Cell Images
> - We revised the introductory section for the Malaria Cell Images dataset to make clearer what the main task is. The main task is detecting the malaria virus (which looks like pink spots in these images).
> - As explained in Section 4.2.2, the classes are indeed unbalanced, therefore, we are using AUC instead of accuracy as a metric.
> - We added the true class label for each cell to Figure 3. We argue that since the reconstructions with only z_d and with only z_y result in circles the information about the original cell shape must be captured by z_x.
>
> 10. Comparison to Domain adversarial training
> - As mentioned in Answer 2. (about domain adversarial training for domain generalization), we argue that domain adversarial training is well established baseline for domain generalization.
> - For 5 out of 10 domains the improvements of DIVA are statistically significant.
>
> [1] - Cai et al., 2019 - Learning Disentangled Semantic Representation for Domain Adaptation
> [2] - Khemakhem et al., 2019 - Variational Autoencoders and Nonlinear ICA: A Unifying Framework
> [3] - Ganin et al., 2015 - Domain-Adversarial Training of Neural Networks
> [4] - Lafarge et a., 2017 - Domain-adversarial neural networks to address the appearance variability of histopathology images

---

> > ### Comment · AnonReviewer3 · 2019-11-14
> > **Thanks for your response.**
> >
> > I would like to thank the authors for their response. Their responses address most of my concerns sufficiently well. However, there are still two points on which I am not very convinced. The first point is the comparison with [ref1]. The current work has almost the same idea (regarding the generative model) with [ref1] which is published in IJCAI this year. I understand some differences of the two works as explained by the authors, but it is necessary to compare with this work to show the superority of the current method. The second point is w.r.t. the variation variable z_x. As shown in the table. (11), in average, the variant without z_x does not differ a lot from the one with z_x. Although using z_x indeeds makes some improvements in some tasks e.g., M_0 and M_75, the performance without z_x also outperforms in some tasks. This results in two questions (1) is z_x reall helpful in the task (I agree with the intuitive interpretation, but the results does not support the intuition well), (2) when should z_x be used since in some tasks it helps, but in some tasks, it may not. With these questions, I am more curious about the comparison with [ref1] since [ref1] does not consider z_x.

---

> > > ### Author Response · Authors · 2019-11-14
> > > **Thanks for the fast reply!**
> > >
> > >
> > > - The generative part of DIVA is modeled after what we believe is the true ground truth generative model for domain generalization datasets like Rotated MNIST and Malaria Cell Images. We argue that especially the qualitative results in Section 4.1.1, 4.1.2 and Appendix 5.1.4 show that each latent space captures a different aspects of x. Furthermore, the new experiments in Table 11 show that there are significant improvements for some tasks, while in all other cases there is no significant difference (overlapping standard error intervals)
> > > - For the completeness of our paper we will run additional experiments comparing DIVA and the model proposed in [ref1]. Judging from the results in Table 11, we do not expect the models to differ a lot in actual classification performance. However, we believe that each model comes with its own advantages. In the case of DIVA, we argue that especially the easily interpretable latent space is a valid contribution. We argue that the comparison of the models should go beyond classification accuracies.
> > > - Unfortunately the implementation of [ref1], hyperparameter search and final evaluation will not be done before the end of the deadline for author comments and responses

---

### Official Review · AnonReviewer2 · 2019-10-24
**Official Blind Review #2**

**Rating:** 3

**Review:**

This work proposes to solve domain generalization problem in a Bayesian way. The idea is relatively simple: use three hidden variables to encode the domain-related, label-related and the residual information from the original signal.

Some questions:
-	My major concern is the intuition for the proposed algorithm. Though the author uses z_d, z_x and z_y to encode different information from x, it is unclear why they are disentangled. It is also possible that all of them share the same information. In other words, why Figure 2 can be derived from this setting is not well explained. I am also confused on why the model can generalize to unseen target. Only using the learned encoder/decoder from training is hard to generalize.
-	For Figure 2, the first column shows z_d. As a common sense, 30 and 45 degree should be more similar than 30 and 60. However, it seems that the cluster center between 30 and 60 is closer than 30 and 45. Is there any justification? My further concern is whether z_d is meaningful at all. What does z_d look like when applying the model on the unseen testing domain?  (Figure 6 should plot in the context with training domains.)
-	 A single k-means method can cluster on rotated MNIST by labels. So I think this property should be kept in the feature without any supervised information. However, z_d seems to remove all label information away. I’m not sure why this can happen.
-	(Minor) The datasets used for comparison are not discriminative. Maybe the encoder structure is more important than the domain generalization method itself. More challenging datasets are expected.

I would like to improve my score if the author can give a reasonable intuition on why the model can generalize on new domains.


**Experience Assessment:**

I have published one or two papers in this area.

**Review Assessment: Checking Correctness Of Derivations And Theory:**

I assessed the sensibility of the derivations and theory.

**Review Assessment: Checking Correctness Of Experiments:**

I assessed the sensibility of the experiments.

**Review Assessment: Thoroughness In Paper Reading:**

I read the paper at least twice and used my best judgement in assessing the paper.

---

> ### Author Response · Authors · 2019-11-13
> **Rebuttal**
>
>
> 1. Disentanglement in DIVA
> - In the VAE framework (on which DIVA is based on) the aggregated posterior is forced to be close to the prior. By partitioning the prior of the latent space into three independent parts z_d, z_x and z_y, DIVA will be forced to learn aggregate posteriors that match those priors, hence disentangling those sources of variation.
> - In addition, very recent work of [1] shows that conditional priors, like p_theta_d(z_d|d) and p_theta_y(z_y|y),  lead to identifiability guarantees in VAEs.  A property that vanilla VAEs are lacking.
> - Last, in Appendix 5.2.5 we show that z_x and z_d are not useful for predicting y
>
> 2. Ordering of domains in Figure 2
> - We updated Figure 6. It shows that DIVA learns a distinct cluster for each training domain. Furthermore the test domain forms its own cluster as well.
> - We argue that the somewhat surprising order of domains in the latent space z_d is due to the individual rotation angle of each MNIST digit in addition to the rotation angle of each domain. As a result, z_x is capturing parts of the rotation angle. [2] came to a similar conclusion.
> - In appendix 5.1.4, we show that we can generate samples conditioned on the domain label. Here we clearly see that different d change the input as expected.
>
> 3. K-means on rotated MNIST
> - Unfortunately, we are not sure what the suggestion is here. Can you please elaborate on this point?
> - The argument why z_d does not capture any label information is the same as in Answer 1. (the argument about disentanglement in DIVA).
>
> 4. Encoder structure
> - For both datasets (Rotated MNIST and Malaria Cell Images) all compared methods use the same encoder architecture. Therefore, we can ensure that the differences in performance do not depend on the encoder architecture. Furthermore, the encoder architecture differs vastly between the two datasets. For Rotated MNIST the LeNet5 architecture is used, while for Malaria Cell Images a larger ResNet architecture is used. A detailed description of the used architectures can be found in the Appendix.
>
>
> [1] - Khemakhem et al., 2019 - Variational Autoencoders and Nonlinear ICA: A Unifying Framework
> [2] - Wang et al., 2019 - Learning Robust Representations by Projecting Superficial Statistics Out

---

### Author Response · Authors · 2019-11-13
**Upload of a revised version**

We would like to thank all reviewers for their time and insightful remarks. We included all of the additional insights and experiments in the revised version of the paper.

---

### Decision · Program_Chairs · 2019-12-19

**Decision:**

Reject

**Comment:**

This paper addresses the problem of domain generalization. The proposed solution, DIVA, introduces a domain invariant variational autoencoder. The latent space can be decomposed into three components: category specific, domain specific, and residual. The authors argue that each component is necessary to capture all relevant information while keeping the latent space interpretable.

This work received mixed scores. Two reviewers recommended weak reject while one reviewer recommended weak accept. There was extensive discussion between the reviewers and authors as well as amongst the reviewers. All reviewers agreed this is an important problem statement and that this work offers a compelling initial approach and experiments for domain generalization. There was disagreement as to whether the contributions as is was sufficient for acceptance. Some reviewers were concerned over similarity to [ref1], this work appears close to the time of ICLR submission and is therefore considered concurrent. However, despite this, there was significant confusion over the proposed solution and whether it is uniquely useful for domain generalization or for other areas like adaptation or transfer learning with reviewers arguing that experiments in these other settings would have helped showcase the benefits of the proposed approach. In addition, there was inconclusive evidence as to whether the two latent components were necessary.

Considering all discussions, reviews, and rebuttals the AC does not recommend this work for acceptance. The contribution and proposed solution needed substantial clarification and the experiments need additional analysis to explain under what conditions each latent component is needed either to improve performance or for interpretability.